# Nanosheet-Facilitated Spray Delivery of dsRNAs Represents a Potential Tool to Control *Rhizoctonia solani* Infection

**DOI:** 10.3390/ijms232112922

**Published:** 2022-10-26

**Authors:** Xijun Chen, Tong Shi, Tao Tang, Chen Chen, You Liang, Shimin Zuo

**Affiliations:** 1College of Horticulture and Plant Protection, Yangzhou University, Yangzhou 225009, China; 2Key Laboratory of Crop Genetics and Physiology of Jiangsu Province/Key Laboratory of Plant Functional Genomics of the Ministry of Education, College of Agriculture, Yangzhou University, Yangzhou 225009, China

**Keywords:** spray-induced gene silencing (SIGS), double-stranded RNA (dsRNA), disease controlling, *Rhizoctonia solani*

## Abstract

*Rhizoctonia solani* is one of the important pathogenic fungi causing several serious crop diseases, such as maize and rice sheath blight. Current methods used to control the disease mainly depend on spraying fungicides because there is no immunity or high resistance available in crops. Spraying double-strand RNA (dsRNA) for induced-gene silencing (SIGS) is a new potentially sustainable and environmentally friendly tool to control plant diseases. Here, we found that fluorescein-labelled *EGFP*-dsRNA could be absorbed by *R. solani* in co-incubation. Furthermore, three dsRNAs, each targeting one of pathogenicity-related genes, *RsPG1*, *RsCATA,* and *RsCRZ1*, significantly downregulated the transcript levels of the target genes after co-incubation, leading to a significant reduction in the pathogenicity of the fungus. Only the spray of *RsCRZ1* dsRNA, but not *RsPG1* or *RsCATA* dsRNA, affected fungal sclerotium formation. dsRNA stability on leaf surfaces and its efficiency in entering leaf cells were significantly improved when dsRNAs were loaded on layered double hydroxide (LDH) nanosheets. Notably, the *RsCRZ1-dsRNA-LDH* approach showed stronger and more lasting effects than using *RsCRZ1-dsRNA* alone in controlling pathogen development. Together, this study provides a new potential method to control crop diseases caused by *R. solani*.

## 1. Introduction

The basidiomycete fungus *Rhizoctonia solani* Kühn [teleomorph *Thanatephorus cucumeris* (Frank) Donk] causes serious diseases such as sheath blight in more than 27 plant families including economically important monocots and dicots, such as rice, wheat, maize, soybeans, potatoes, melons, tobacco, and beets [1,2,3]. For more than a decade, sheath blight caused by *R. solani* has been the most serious disease of rice in China, measured by both the diseased area and yield loss [4]. The disease has also become one of the most serious rice diseases in many other countries, causing huge yield losses [5,6]. On maize, banded leaf and sheath blight caused by *R. solani* led to losses in grain yield to the extent of over 90 percent [7]. Although various methods including agronomic practices, biological control, and chemical agents have been used to control the disease [8,9,10,11,12], the situation is far from satisfactory due to the consequential environmental problems from chemical fungicides and the inefficiency of other control methods [13,14,15,16,17,18]. Therefore, it is urgently needed to develop potential new methods to control the disease.

Foreign double-stranded RNAs (dsRNAs) can inhibit the corresponding gene activity based on homology [19]. dsRNAs were found to show potent and specific interference effects and could be delivered from plants to their pathogens through cross-kingdom transportation [20,21,22,23]. Furthermore, recent studies showed that a dsRNA spraying method called spray-induced gene silencing (SIGS) for controlling pathogen infections in plants represents an attractive alternative strategy [24,25]. Direct dsRNA (*ATC*, *RP_16S*, and *GCS_H*) sprays on detached soybean leaves reduced *Phakopsora pachyrhizi* biomass and its gene expression levels, which went about from 17.0~25.1% and 28.2~32.9% in the detached soybean leaves sprayed with water, respectively [26]. The foliar application of dsRNAs targeting genes such as SS1G_6487 and SS1G_01703 in *Sclerotinia sclerotiorum* resulted in significant reductions of up to 85% in lesion size [27]. Although most plants and pathogens could take up exogenous dsRNAs, different pathogen and plant species had different capacities for dsRNA uptake [28,29]. Layered double hydroxide (LDH), a type of non-toxic, degradable, designer nanosheet, can efficiently carry dsRNAs and promote the uptake of dsRNAs in plant leaves. When CMV2b-dsRNA-Cy3 and CMV2b-dsRNA-Cy3-LDH were applied to *Arabidopsis thaliana* leaves individually, and leaves were rinsed rigorously, most of the CMV2b-dsRNA-Cy3 was washed away, whereas CMV2b-dsRNA-Cy3-LDH largely remained on leaves after rinsing [28]. With regard to the application of controlling *R. solani*, Qiao et al. [29] found that the application of *Rs-DCTN1+SAC1*-dsRNA and *Rs-PG*-dsRNA targeting *R. solani DCTN1*, *SAC1,* and *PG* genes resulted in a reduction in disease symptoms and fungal biomass. However, effective control of *R. solani* via SIGS remains to be extensively investigated.

Plants have evolved a wide range of defense strategies, including pattern-triggered immunity (PTI) and effector-triggered immunity (ETI) as the most important immune signaling strategies, to counter pathogen attacks [30,31,32]. The two branches of immune signaling were both associated with the production of reactive oxygen species (ROS), including superoxide and hydrogen peroxide (H_2_O_2_), which can kill pathogens and further activate immune responses [30,33]. Likewise, pathogens have evolved strategies to overcome host defenses, such as overexpression of catalases (CAT) to counteract pro-oxidants generated by plants to increase its pathogenicity [34,35]. In oxidative environments, ROS-degrading enzymes such as catalases play crucial roles for the survival of pathogens. Knockout of the gene encoding a catalase increases the sensitivity to exogenously added H_2_O_2_ and delayed *Magnaporthe oryzae* infection at early stages [36]. Calcineurin responsive zinc finger 1 (CRZ1), a well-characterized target protein of calcineurin, is a transcription factor which plays a critical role in diverse cellular processes, such as cellular function regulation, tolerance to stress conditions, and virulence of fungi [37,38,39]. When *CRZ1* was mutated, the virulence of several fungi, such as *Pyricularia oryzae*, *Metarhizium acridum*, and *Crytococcus deneoformans*, was reduced [40,41]. In *Alternaria alternate*, AaCrz1 was required for infection structure differentiation; *∆AaCrz1* was impaired in vegetative growth, spore morphology, cell wall integrity, and virulence [42]. CRZ1 even has been considered a promising therapeutic target in echinocandin-resistant *Candida glabrata*, which kills more than 1.6 million patients each year worldwide [43]. Polygalacturonase (PG) is an important pathogenic factor secreted by most pathogens, including *R. solani*. PG cleaves pectin, one of the major plant cell wall components, causes cell separation, and macerates host tissues [44,45]. In many fungi, PGs are required for full virulence [46]. *Claviceps purpurea* mutants lacking *CpPG1* and *CpPG2* were not affected in their vegetative properties, but their virulence to rye was almost completely lost [47].

Previous reports have shown that *CAT* and *PG* play critical roles in the pathogenicity of *R. solani* [44,45,48,49]. In order to determine whether exogenous sprays of dsRNAs targeting *RsPG1*, *RsCAT1,* and *RsCRZ1* can reduce the virulence of *R. solani* and its destructive effects on host crops, we evaluated the effects of the SIGS technology on controlling *R. solani* infection using dsRNAs specific to these genes. We found that dsRNAs loaded on LDH could significantly extend their inhibitory effects on *R. solani* in maize. Our study provides a new potential method to control crop diseases caused by *R. solani*.

## 2. Results

### 2.1. R. solani Is Able to Uptake Exogenous dsRNA

To determine whether *R. solani* can take up exogenous dsRNA, we co-incubated mycelia with fluorescein-labelled *EGFP*-dsRNA in the PDB culture. At 24 h after co-incubation (HAC), we detected fluorescence using a microscope and observed intense fluorescence signals in the mycelia treated with fluorescein-labelled *EGFP*-dsRNA, while little autofluorescence was found in the mock treatment (Figure 1A). To further confirm that the dsRNA entered cytoplasm of the fungal cells, we extracted the protoplasts of the mycelia co-incubated with fluorescein-labelled *EGFP*-dsRNA. Under fluorescence confocal microscopy, we found that the fluorescence signals were clearly located in *R. solani* protoplasts after MNase treatment (Figure 1B). Four days after co-incubation, macroscopic sclerotia could be found easily in the PDB plate. Notably, we found that fluorescence signals could also be observed in the mycelia that formed sclerotia, although they were much weaker than those at 24 HAC (Figure 1C). 

To determine whether the *EGFP*-dsRNA affects the growth of *R. solani*, we inoculated mycelial plugs on sterilized cellophanes on PDA medium and sprayed *EGPF*-dsRNA on them. After measuring colony diameter and mycelial weight, we found no significant differences between the treatment and the control (Figure 1D,E). Together, these data demonstrate that *R. solani* is capable of taking up exogenous dsRNA during mycelial development.

### 2.2. Transcript Levels of Pathogenicity-Related Genes can Be Significantly Reduced by the Treatment of Corresponding dsRNAs

To determine the effects of dsRNAs on regulating endogenous gene expression, we selected three pathogenicity-related genes, *RsPG1*, *RsCAT,* and *RsCRZ1*, as potential interference targets. *RsPG1*-dsRNA, *RsCAT*-dsRNA, and *RsCRZ1*-dsRNA from nonconserved regions of each gene were generated and individually co-incubated with mycelia on PDB culture. We performed RT-qPCR and found that each of the three target genes in the pathogen were significantly downregulated at 24 HAC: all reduced to approximately 30% compared with the control (Figure 2A). The deletion of *CRZ1* in fungi has previously been found to cause high sensitivity to Ca^2+^ and Mg^2+^ [50]. We also found that *R. solani* co-incubated with *RsCRZ1*-dsRNA caused high sensitivity to both Ca^2+^ and Mg^2+^, and the sensitivity was positively correlated with the concentration of both ions (Figure 2B). Previous reports also showed that *CRZ1* in fungi regulates the expression of several downstream genes, including *NCS, MAPK*, *CYP51,* and *CaMK,* to affect pathogenicity [51,52]. We subsequently detected the RNA levels of these genes in *R. solani* and found that the reduction in *RsCRZ1* only affected *RsNC*S RNA expression levels (Appendix A). Together, we conclude that co-incubation with dsRNA can significantly reduce the expressions of the three pathogenicity-related genes. 

We also measured mycelial growth characteristics including colony diameter and mycelium weight for each dsRNA treatment. We found no significant differences in the two traits between the dsRNA treatments and the control, indicating that these three dsRNAs do not affect mycelial development (Figure 2C,D). Interestingly, we found that the pathogens treated with *RsCAT1*-dsRNA produced sclerotia much faster and more than the control at 3 HAC (Figure 2E), while there were no differences for treatments with the other two dsRNAs. These results indicate that treatments with these three dsRNAs have little influence on mycelial development.

### 2.3. Treatments with Pathogenic Gene-Derived dsRNAs Significantly Reduced R. solani Virulence

To test the effects of the three pathogenic gene-derived dsRNAs on *R. solani* virulence, maize leaves were inoculated with the pathogen 24 h after treatment with individual dsRNA. Three days after inoculation (DAI), we found that the maize leaves treated with each of the three dsRNAs produced clearly smaller diseased areas compared with the control, with lesion areas reduced by 51.8%, 67.9%, and 71.4% for *RsCAT*-dsRNA, *RsCRZ1*-dsRNA, and *RsPG1*-dsRNA, respectively (Figure 3A). We detected the biomass of *R. solani* in the diseased leaves by quantitative PCR and found that the dsRNA treatments all significantly suppressed growth of the pathogen, reflected by an over 50% reduction in biomass (Figure 3B). We further detected *RsPG1*, *RsCAT,* and *RsCRZ1* RNA levels in the pathogen from the diseased leaf parts and found that the dsRNA treatments significantly reduced the expressions of *RsPG1*, *RsCAT,* and *RsCRZ1* to about 31.5%, 43.3%, and 35.7% (Figure 3C). 

Catalase (CAT) is responsible for removing H_2_O_2_ from cells. To verify whether *RsCAT*-dsRNA can affect H_2_O_2_ accumulation at the infection sites, we sprayed *RsCAT*-dsRNA on the surface of tobacco leaves 24 h before the pathogen was inoculated. RT-qPCR data showed that the *RsCAT* gene, but not the *NbCAT* gene, in tobacco was apparently downregulated (Figure 3C, Appendix A), which reflects the fact that the *NbCAT* and *RsCAT* genes share very low homologies (Appendix A). Subsequently, we detected the sensitivity of the *RsCAT*-dsRNA-treated *R. solani* to H_2_O_2_ on medium and confirmed that the *RsCAT*-dsRNA-treated pathogen showed a higher sensitivity to H_2_O_2_ than the untreated, reflected by the significantly smaller colony diameters on plates with 0.01 M or 0.1 M H_2_O_2_ (Figure 3D). Furthermore, we found that the *RsCAT*-dsRNA-treated *R. solani* clearly led to more H_2_O_2_ accumulation than the untreated on tobacco leaves 12 h post inoculation (HPI) (Figure 3E). This result shows that *RsCAT*-dsRNA inhibits RsCAT function on scavenging H_2_O_2_ and ultimately reduces *R. solani* pathogenicity. Together, these data indicate that externally applied pathogenic gene-derived dsRNAs can significantly reduce *R. solani* pathogenicity.

### 2.4. Maize Leaves Are Able to Take up Exogenous dsRNA

In addition to the fact that dsRNA is absorbed by *R. solani*, dsRNA can also be taken up by plants when sprayed on plant surfaces, which may further enhance the efficiency of dsRNA on disease control [53]. In order to explore the absorption and transfer capacity of dsRNA in maize, we sprayed fluorescein-labeled *EGFP*-dsRNA onto wounded maize leaves. Results showed that fluorescein-labeled *EGFP*-dsRNA was absorbed by maize leaves and was absorbed more efficiently via wounds than healthy epidermis (Figure 4A). Strong fluorescence signals extended from the cut to the distal section after *EGFP*-dsRNA was sprayed on wounded maize leaves, whereas only faint fluorescence signals were detected in intact leaves under the same conditions (Figure 4A). The results of the RT-qPCR showed that the relative amount of the *EGFP*-dsRNA at the wounded site was about five times higher than at the intact site (Figure 4B). 

Moreover, the relative amount of the *EGFP*-dsRNA at the local site was about three times higher than at distant sites (Figure 4C). We further detected fluorescence signals in tissues far from the cut site and observed strong fluorescence signals from epidermal cells to vascular cells, including phloem and xylem cells (Figure 4D), implying that dsRNA was mainly transported through the vascular system. However, without a continuous supply of *EGFP*-dsRNA, exogenous *EGFP*-dsRNA almost disappeared on maize leaf surfaces 72 h after treatment (Appendix A). Together, these data indicate that the dsRNA can be taken up by maize, although it cannot stay stable on the surface for a long time. 

### 2.5. LDH Improves dsRNA Stability on Leaf Surface, Enhancing Control of Maize Sheath Blight 

In order to increase the stability of dsRNA on plant surfaces, we employed layered double hydroxide (LDH) clay nanosheets as nanocarriers for dsRNAs before spraying on plant surfaces. Previous studies have shown that the loading efficiencies at different dsRNA:LDH mass ratios were significantly different [54,55]. Therefore, we first measured the loading efficiency of *EGFP*-dsRNA with LDH and found that the loading efficiency increased gradually when the mass ratio of dsRNA:LDH decreased from 1:1 to 1:16. At the ratio of 1:1, only about 5% of the *EGFP*-dsRNA was loaded on the LDH, while at 1:16, the *EGFP*-dsRNA was completely loaded on the LDH, as no *EGFP*-dsRNA migrated during electrophoresis (Figure 5A). In addition, we found that increasing the co-incubation time improved the loading efficiency: while only about 5% of the dsRNA was loaded at the ratio of 1:1 when co-incubated for half an hour, most of the dsRNA was loaded at a 1:1 ratio when co-incubated for 2 h, and almost all of the dsRNA was loaded at ratios lower than 1:2 when co-incubated for two hours (Figure 5B). To assess the efficiency of dsRNA transfer to plant cells facilitated by LDH, fluorescein-labeled *EFGP*-dsRNA loaded on LDH, *EFGP*-dsRNA, and LDH were individually sprayed on tobacco and maize leaf surfaces. After 24 hours of incubation, the leaves were rinsed under strong running water for 1 min, and then observed for fluorescence under a confocal microscope. The tobacco leaves sprayed with *EGFP*-dsRNA-LDH showed strong fluorescent signals, while no fluorescent signals were observed for those sprayed with LDH and very weak fluorescence for those sprayed only with *EGFP*-dsRNA (Figure 5C). The same trend was found on maize leaves. Moreover, observations of the sectioned samples showed that *EGFP*-dsRNA-LDH was absorbed by the epidermal cells of the maize leaves and was transported to the vascular cells through intercellular space (Figure 5D). 

With LDH, we further tested the effect of *RsCRZ1*-dsRNA-LDH on controlling maize sheath blight. We found that the lesion areas on *RsCRZ1*-dsRNA-LDH-treated leaves were clearly smaller than *RsCRZ1*-dsRNA-treated and water-treated leaves at all three time points after inoculation (Figure 5E). Comparatively, *RsCRZ1*-dsRNA only showed significant effects on suppressing disease development at 3 DPI, but not at 7 DPI or 20 DPI. In contrast, *RsCRZ1*-dsRNA-LDH showed a significantly durable effect on suppressing disease development, reducing lesion areas from 30% to 47% at all three time points compared to *RsCRZ1*-dsRNA and water (Figure 5F). We further detected the *CRZ1* mRNA levels in *R. solani* by RT-qPCR and found that the *CRZ1* mRNA levels were clearly lower in *RsCRZ1*-dsRNA-LDH treatment than in *RsCRZ1*-dsRNA treatment at all three time points (Figure 5G). At 20 DPI, almost no *CRZ1* mRNA was detected in *RsCRZ1*-dsRNA-LDH treatment. These results indicate that LDH helps dsRNA to enter plant cells and extend the protection time of Rs*CRZ1*-dsRNA on controlling maize sheath blight.

## 3. Discussion

### 3.1. SIGS Presents a Promising Alternative Method to Control Diseases Caused by R. solani

*R. solani* is the causal agent of a wide range of crops, causing enormous losses. However, due to the complex necrotrophic nature of the fungus, it is extremely difficult to efficiently control the disease. Ever since dsRNA was found to inhibit the function of its cognate gene based on homology, technologies of host-induced gene silencing (HIGS), virus-induced gene silencing (VIGS), and filamentous organism-induced gene silencing (FIGS) were developed and used to control plant diseases [19,56,57,58,59]. With regard to *R. solani*, Ghosh et al. [60] generated transgenic tomato plants harboring a dsRNA specific to the *RsCRZ1* fungal gene and found that the transgenic plants exhibited significant reductions in disease symptoms and colonization depth of the pathogen. When RNA interference constructs of 33 candidate pathogenicity genes from *R. solani* were infiltrated into *Nicotiana benthamiana* leaves, 29 resulted in a significant reduction in necrosis caused by the pathogen [61]. In addition, virus-induced gene silencing of the *NUOR* gene in tomatoes reduced susceptibility to *R. solani* [62]. Unfortunately, all of these technologies require genetic manipulation, but these products are currently banned in Europe and resisted by many consumers [63,64]. Therefore, an innovative, effective, operable, and nongenetically modifying method for plant protection is urgently needed. 

Koch et al. [24] found that exogenous dsRNAs could be efficiently taken up from the environment by fungal pathogens, and then processed into small RNAs (sRNAs), which silence pathogen genes with complementary sequences. Externally spraying dsRNAs targeting pathogenicity-related genes on the surface of hosts significantly inhibits diseases [29,53,65]. Although previous studies indicated that using *RsDCTN1+SAC1-dsRNA* or *RsPG-dsRNA* to target *R. solani* genes could protect plants from *R. solani* infection [29], the SIGS technology on controlling *R. solani* remains largely unknown.

To systematically test if SIGS is able to control *R. solani* infection, we first confirmed that *R. solani* could take up exogenous dsRNA by co-incubating the fungus with fluorescein-labelled *EGFP*-dsRNA (Figure 1). Then, we further tested if dsRNA derived from pathogenic genes could suppress *R. solani* pathogenicity. When each of the three dsRNAs derived from the three pathogenic genes of *R. solani* was co-incubated with the pathogen, the expression level of the corresponding gene was significantly reduced (Figure 2A). In consistency, *R. solani* pathogenicity-related biological characteristics, such as sensitivity to metal ions and H_2_O_2_, time, and number of sclerotia formations of *R. solani*, were significantly changed (Figure 2B–E and Figure 3D). More importantly, spraying dsRNAs on maize leaves significantly reduced the expression levels of the target genes, pathogen biomass on the leaves, and lesion sizes, suggesting the applicability of this approach (Figure 3A–C). One question remains unanswered: why the three different dsRNAs have very similar levels of effects on reducing endogenous gene expressions and lesion sizes. It is worth further study to determine whether the SIGS strategy, using different single dsRNAs, can only reach this level, and if combining the three dsRNAs can significantly boost the effects higher than using individual dsRNAs on disease control. However, although many questions in this field remain to be investigated, our data clearly suggest that SIGS using pathogenic gene-derived dsRNAs serves as a potential strategy to control *R. solani* diseases.

### 3.2. Layered Double Hydroxide (LDH) Significantly Increases the Effects of SIGS on Disease Control 

When using SIGS to control plant diseases in the field, the critical issues are the efficiency of dsRNA absorbed by plant hosts and the activity of dsRNA maintained on leaf surfaces. Although dsRNA is not as easily degraded as single-stranded RNA (ssRNA), it can be inactivated because of a large amount of RNA enzymes inside and outside of cells in the environment [66,67]. The textural features of nanoporous materials, including high surface area, large pore volume, tunable pore size, and stability in different environments, confer a more efficient loading of pesticides [68]. Moreover, it can target the delivery of pesticides to achieve increased disease control efficiencies over longer durations and to prevent degradation of the encapsulated molecules under adverse environmental conditions [69,70]. If nanoparticles could be used to deliver and protect dsRNAs, it will effectively improve dsRNA application prospects in the field. Although the BioDirect technology from Monsanto as an RNAi application has been reported, details of this technology have not been available. Previous studies reported that the LDH nanomaterial can effectively load dsRNAs and affords longer virus protection to sprayed leaves and newly emerged unsprayed leaves when dsRNA-LDH was topically applied [28]. Here, we first tested the dsRNA loading efficiency of LDH and found that LDH could effectively load dsRNA in two hours at a 1:2 (dsRNA:LDH) mass ratio (Figure 5B). When dsRNA-LDH was sprayed on plant leaves, the efficiency of dsRNA entry and transport in plant tissues was greatly improved, and the protection duration was significantly extended compared to spraying dsRNA alone (Figure 5C–G). This means that LDH is an ideal microcarrier to extend the effects of dsRNA on disease control. Therefore, it is highly worthwhile to further test the effects of the three pathogenic gene-derived dsRNAs loaded on LDH individually and the effect of combining all three dsRNAs on disease control in fields. 

## 4. Materials and Methods

### 4.1. Strain and Plant Materials

*The Rhizoctonia solani* AG1-IA strain YN-7 was isolated from diseased rice plants in the Jiangsu province, China, and identified by morphology and molecular biology. Genomic DNA of YN-7 was extracted with a DNA extraction kit (Solarbio, Beijing, China) and used to amplify the ITS region with the common primers ITS4 (5′-GGAAGTAAAAGTCGTAACAAGG-3′) and ITS5 (5′-TCCTCCGCTTATTGATATGC-3´). The anastomosis group of the strain was determined based on its 28S rDNA sequence which was amplified with the specific primer pair for AG 1-IA (*R. solani* AG common primer: 5′-CTCAAACAGGCATGCTC-3′, *R. solani* AG 1-IA specific primer: 5′-CAGCAATAGTTGGTGGA-3′). Seedlings of *Nicotiana benthamiana* and *Zea mays* were grown in a plant growth chamber at 25 °C, 60% relative humidity, and a 12 h day/night cycle.

### 4.2. Synthesis of dsRNA In Vitro

About 200 to 300 nucleotides (nt) of the target genes were selected for dsRNA synthesis. Following the instructions of the T7 RNAi Transcription Kit (Vazyme, Nanjing, Jiangsu, China), primers containing the T7 promoter sequence at both the 5′ and 3′ ends were designed for dsRNA synthesis by PCR. After purification, the DNA fragments containing the T7 promoter were used for in vitro transcription. Primers used for synthesizing dsRNAs are listed in Appendix A. To synthesize the fluorescein-labelled dsRNA in vitro, a fluorescein RNA labeling mix (Roche, Mannheim, Germany) was used for the synthesis following the manufacturer’s instructions.

### 4.3. Fluorescein-Labelled dsRNA Uptake by R. solani

A mycelial plug of *R. solani*, 5 mm in diameter, was placed in the center of a sterile cellophane, placed on the PDB culture medium, and sprayed with 100 μL of 50 ng/μL fluorescein-labelled dsRNA. After being incubated at 28 °C for 24 h, mycelia were collected and treated with a KCl buffer or 75 U of micrococcal nuclease enzyme dissolved in a KCl buffer to degrade the dsRNA on the surface of the mycelium at 37 °C for 30 min. A part of the mycelium was used to observe and take graphs, and the remaining was used to prepare protoplasts. Fresh mycelium was collected, washed with sterile water twice, slightly press-dried to remove residual liquid, and transferred into an enzymatic solution (2.5 g of lysozyme and 0.5 g of driselase dissolved in 10 mL of 0.6 M mannitol and filtered through a 0.22 μm microporous membrane). After incubation at 28 °C at 80 rpm for 3 h, the product of enzymatic hydrolysis was filtered through a 200-mesh cell strainer, and the filtrate was centrifuged at 4 °C at 4000 rpm for 5 min. The supernatant was discarded, and the precipitate was washed twice with 0.6 M mannitol to obtain protoplasts. The fluorescent signals of the mycelia and protoplasts were detected using an Olympus BX 51 confocal microscope. Some mycelial plugs inoculated on the cellophane were cultured at 28 °C for 4 days, and sclerotia were collected, sealed with wax, and used for slicing. The slices were used for photo taking under a confocal microscope.

### 4.4. R. solani Growth Rate Determination

To determine the effect on the growth of *R. solani* after dsRNA uptake, one hundred microliters of dsRNA at 50 ng/μL was spread evenly on a PDA medium surface, and plugs 5 mm in diameter were placed in the center of the PDA culture medium in petri dishes, which then were placed in a constant temperature incubator at 28 °C. The growth and development status of the fungus was observed continuously. When the control colony was about to overgrow the plate, the colony diameter of the fungus in each dish was determined. After being incubated at 28 °C for 7 days, sclerotia in the dishes were collected and weighed. Three petri dishes were used as three replicates, and the experiment was repeated twice.

### 4.5. RNA Extraction and Reverse Transcription Quantitative PCR (RT-qPCR)

Total RNAs from the fungus and plant leaves were extracted using the RNA-easy isolation reagent (Vazyme, Nanjing, Jiangsu, China) and purified with DNase I (Vazyme, Nanjing, Jiangsu, China). First-strand cDNA was synthesized using the S*EasyScript*^®^ All-in-One First-Strand cDNA Synthesis SuperMix (Transgene, Beijing, China). RT-qPCRs were performed using the CFX96 (Bio-Rad, Hercules, CA, USA) Connect system using SYBR Green mix (Transgene, Beijing, China) with the following program: 94 °C for 30 s, followed by 40 cycles of 94 °C for 5 s and 60 °C for 30 s. RNA expression levels were calculated using the ΔΔC_t_ method. The primers used for RT-qPCRs are listed in Appendix A. Each treatment included three replicates, and the experiment was repeated three times.

### 4.6. Interference Effect of Exogenous dsRNA on Genes in R. solani

The PDA media in the petri dishes were covered with a sterile cellophane, and 100 microliters of dsRNA at 50 ng/µL was sprayed in each petri dish. Mycelial plugs 5 mm in diameter from the colony edge of the *R. solani* cultured in PDA medium for 36 h were placed in the center of the cellophane. After being cultured at 28 °C for 2 days, mycelia were collected from the cellophanes and used to extract RNA, which was then used to detect the expression levels of the genes in *R. solani*; the *RsGAPDH* gene was used as a reference gene. Each treatment included three replicates, and the experiment was repeated three times.

### 4.7. External Application of dsRNA on the Surface of Plant Material

Synthetic dsRNAs diluted with RNase-free water at a final concentration of 50 ng/μL were sprayed on the surfaces of the leaves of four-leaf stage *Nicotiana benthamiana* and four-week-old *Zea mays* plants. For comparing the uptake abilities of intact and wounded leaves, some maize leaves were wounded with a single-sided blade, and then fluorescein-labelled *EGFP*-dsRNA was sprayed. After 24 h, the leaves were cut from the plants and washed with sterile water before being used to observe with an inverted fluorescence microscope. To determine the protective effects of dsRNAs on plants, dsRNAs targeting different genes including *RsPG*1, *RsCAT,* and *RsCRZ1* were sprayed on the surfaces of maize individually. After a dsRNA had been absorbed by plant cells 24 h post-spray, mycelial plugs 5 mm in diameter were inoculated on the leaves, and the plants were grown in chambers at 28 °C with a 12 h day/night cycle. Lesion areas were measured 3 days post-inoculation (dpi). Five leaves were used in each replicate, and each treatment included three replicates. The experiment was repeated three times.

### 4.8. Detection of the Hydrogen Peroxide or Metal Ion Sensitivity of R. solani 

For the determination of the sensitivity of *R. solani* treated with *RsCAT*-dsRNA to hydrogen peroxide, mycelial plugs 5 mm in diameter from the colony edges on the PDA medium harboring *RsCAT*-dsRNA were placed in the center of the PDA medium with final H_2_O_2_ concentrations of 0.1 mol/L or 0.01 mol/L. In parallel, mycelial plugs from the colony growing on PDA harboring *RsCRZ1*-dsRNA were inoculated in the center of the medium with CaCl_2_ of 0.1 M or 0.2 M and MgCl_2_ of 0.1 M or 0.2 M, respectively. Petri dishes were placed in an electrothermal constant temperature incubator at 28 °C. Colony diameters were measured when the fastest growing colonies were close to overgrowing the plate. Three petri dishes were used as three replicates, and the experiment was repeated twice.

### 4.9. Leaf Staining

To visualize ROS accumulation in tobacco leaves treated with *RsCAT*-dsRNA and inoculated with *R. solani*, leaves were collected 6 h and 12 h after inoculation and stained with a diaminobenzene (DAB) solution at 1.0 DAB mg/mL (prepared in 50 mM tris acetate buffer, pH 5.0) at 25 °C for 24 h in the dark, and then de-stained overnight in an ethanol/acetone solution at a 3:1 ratio. Because of DAB polymerization, H_2_O_2_ was visualized as brown-colored patches on the de-stained leaves. Leaves infected by *R. solani* were incubated in a 0.1% solution of DAB for 2 h, and then boiled in water at 95 °C for 10 min. When the decolorization was complete, the leaves were photographed. Five leaves were used in each replicate, and each treatment included three replicates. All of these experiments were repeated three times. 

### 4.10. DsRNA Loading on LDH and Transportation Facilitated by LDH

To find the optimal and complete loading of dsRNA on LDH, the ratio of *EGFP*-dsRNA to LDH was assessed at 1:1, 1:2, 1:3, 1:4, 1:5, 1:8, and 1:16 by mixing at room temperature for 30 min or 2 h with gentle orbital agitation. Ten microliters each of the *EGFP*-dsRNA/LDH complexes were loaded into the well of a 1.0% agarose gel and electrophoresed to assess the loading ratios. Once loaded onto LDH, dsRNA cannot migrate from the well during electrophoresis. In order to determine whether LDH can facilitate the delivery of dsRNAs into plant leaves, fluorescein-labeled *EGFP*-dsRNA, *EGFP*-dsRNA/LDH, and LDH were sprayed on tobacco and maize leaves individually. After incubation for 24 h at room temperature in the dark, tobacco leaves were rinsed by vigorous pipetting for 2 min. Similarly, maize leaves were washed, sliced, and observed for fluorescein signals. The experiment was repeated twice.

### 4.11. Systemic Protection Assays

After two hours of co-incubation with *RsCRZ1*-dsRNA and LDH at a 1:1 ratio, *RsCRZ1*-dsRNA-LDH was assessed for RNAi-mediated protection on plants compared to naked *RsCRZ1*-dsRNA. Twenty hours after the spray, mycelial plugs were inoculated on maize leaves, and lesion sizes were measured at 3, 7, and 20 days after inoculation.

### 4.12. Statistical Analysis

All data were analyzed using the one-factor ANOVA procedure in the SPSS 22.0 program (IBM, Chicago, IL, USA). Mean comparisons (LSD) were used to determine differences among treatments, where ns, *, and ** mean no significant differences, and *p*-values less than 0.05 and 0.01, respectively.

## 5. Conclusions

SIGS, based on RNAi, is a new potential tool to control plant diseases without genetic manipulation and environmental concerns. When dsRNA targeting genes related the growth, development, and pathogenicity of *R. solani* was co-incubated with the pathogen or spayed on the plant leaves inoculated with the pathogen later, the expression levels of related genes in the pathogen significantly downregulated, and the pathogenicity of the fungus reduced. LDH nanosheets could load dsRNA efficiently, and the dsRNA-LDH could provide stronger and more lasting effects than using naked dsRNA in controlling pathogen development and lesion extension. In short, this study provides a new potential method to control crop diseases caused by *R. solani*.

## Figures and Tables

**Figure 1 ijms-23-12922-f001:**
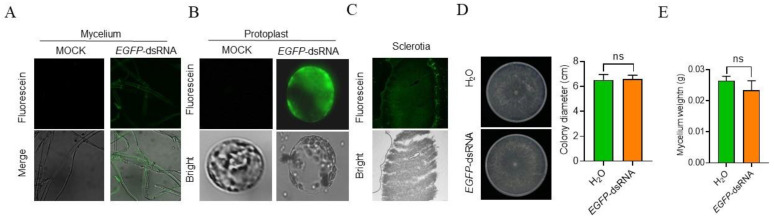
*EGFP*-dsRNA uptake ability of *R. solani*. (**A**) Mycelial blocks of *R. solani* were inoculated on a sterile cellophane placed on potato dextrose agar medium and sprayed with fluorescein-labeled *EGFP*-dsRNA. Strong fluorescent signals were observed in the mycelia growing on the cellophane 24 h later. (**B**) Mycelia collected from the cellophane were treated with a mixture of lywallzyme and driselase for 3 h, and the protoplasts were collected for observation under a confocal microscope. Fluorescein signals inside the protoplasts meant that dsRNA had entered *R. solani* cells. (**C**) Mycelia on the cellophane formed sclerotia 4 days after inoculation. Fluorescein signals are also observed in the inner cells of the sclerotia. (**D**) Colony of *R. solani* in the sterilized cellophanes placed on PDA medium and sprayed with dsRNA. (**E**) Weight of mycelia collected from the sterilized cellophane. ns: no significant differences.

**Figure 2 ijms-23-12922-f002:**
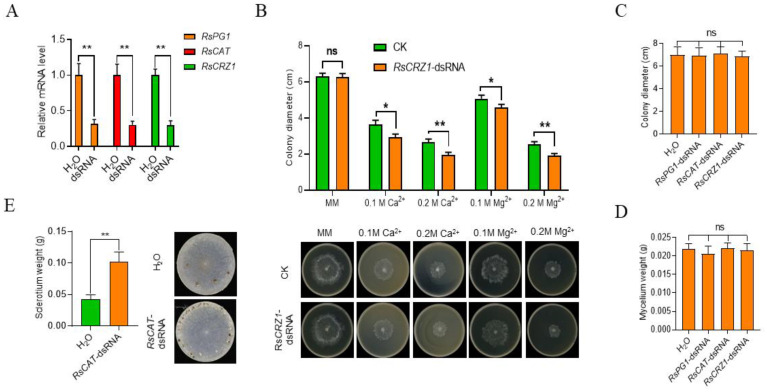
Effects of dsRNAs that target *R. solani* genes on *R. solani* growth and development. (**A**) Expression levels of *RsPG1*, *RsCAT,* and *RsCRZ1* 30 h after spray with dsRNAs targeting these genes on *R. solani* growing on PDA medium. (**B**) Sensitivity of *R. solani* treated with *RsCRZ1*-dsRNA to different concentrations of Ca^2+^ and Mg^2+^. CK, control; MM, minimal medium. (**C**) Colony diameter of *R. solani* co-incubated with *RsPG1*-dsRNA, *RsCAT*-dsRNA, or *RsCRZ1*-dsRNA for 30 h. (**D**) Weight of mycelia collected from the cellophane sprayed with *RsPG1*-dsRNA, *RsCAT*-dsRNA, or *RsCRZ1*-dsRNA. (**E**) Sclerotia production and its weight after *RsCAT*-dsRNA treatment for 7 days on the pathogen in a petri dish. ns, no significant differences; *, *p* < 0.05; **, *p* < 0.01.

**Figure 3 ijms-23-12922-f003:**
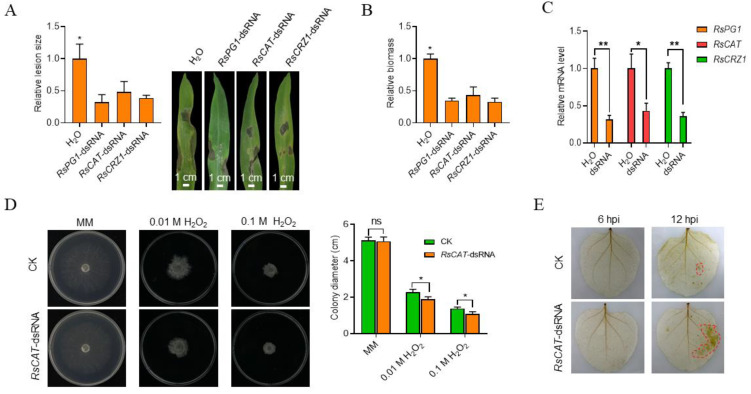
Exogenous spray of dsRNAs targeting pathogenic genes of *R. solani* significantly reduces its virulence. (**A**) Lesion sizes of maize leaves inoculated with *R. solani* mycelial blocks after treatment with dsRNAs targeting the *RsPG1*, *RsCAT,* or *RsCRZ1* gene. Lesion sizes were measured using ImageJ 3 dpi. (**B**) Biomass of *R. solani* in maize leaves 3 dpi. (**C**) Expression levels of the *RsPG1*, *RsCAT,* and *RsCRZ1* genes of *R. solani* 3 dpi in the maize leaves sprayed with different dsRNAs. (**D**) Sensitivity of *R. solani* treated with *RsCAT*-dsRNA to different concentrations of H_2_O_2_. CK, control; MM, minimal medium. (**E**) ROS content in the tobacco leaves sprayed with *RsCAT*-dsRNA. Leaves were taken at 6 HPI and 12 HPI with the pathogen and stained with DAB. ns, no significant differences; *, *p* < 0.05; **, *p* < 0.01.

**Figure 4 ijms-23-12922-f004:**
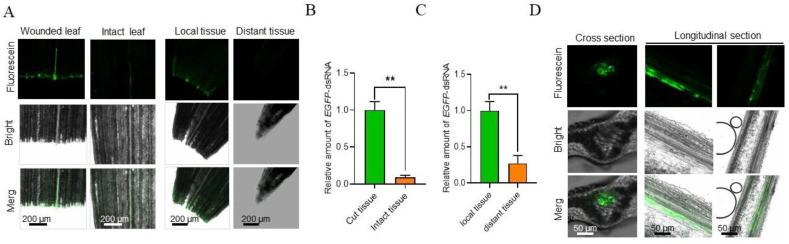
DsRNA uptake by maize leaves and transportation in the vascular system. (**A**) Fluorescein-labeled *EGFP*-dsRNA uptake by wounded and intact maize leaves. Detection of fluorescein signals in leaves 24 h after dsRNA spraying. (**B**) Amounts of *EGFP*-dsRNA in the wounded and intact maize leaves 24 h after spraying. (**C**) Amounts of *EGFP*-dsRNA in local and distant tissues of maize leaves 24 h after spraying. (**D**) Transportation routes of *EGFP*-dsRNA in maize leaves. Leaf cross sections were examined for fluorescein signals in the vascular system, including the phloem and xylem cells. Longitudinal sections of leaves for observation of fluorescein signals in transportation routes: from epidermal cells to vascular cells and from local to distant sites in the vascular system. **, *p* < 0.01.

**Figure 5 ijms-23-12922-f005:**
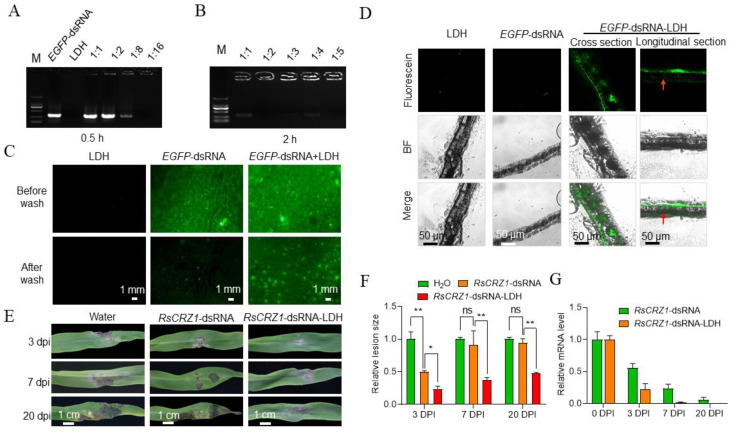
DsRNA loading capacity of LDH and protective effects of *Rs*CRZ1-dsRNA-LDH on plants. (**A**) Electrophoretogram of the mixtures of dsRNA and LDH at different ratios. dsRNA and LDH were mixed at 1:1, 1:2, 1:8, and 1:16 ratios for half an hour. (**B**) Electrophoretogram of the mixtures of dsRNA and LDH at different ratios. dsRNA and LDH were mixed at 1:1, 1:2, 1:3, 1:4, and 1:5 ratios for two hours. (**C**) Effects of LDH on the entry of dsRNA into tobacco leaves. Fluorescent signals were observed under confocal microscopy on tobacco leaves 24 h post-dsRNA treatment and before or after washing. (**D**) Effects of LDH on dsRNA entry into maize leaves and transport inside leaves. Twenty-four hours after *EGFP*-dsRNA-LDH was sprayed on maize leaves, cross and longitudinal sections were observed for fluorescence in the cells and intercellular space. (**E**) Effects of *RsCRZ1*-dsRNA-LDH spray on *R. solani*-caused lesions on maize leaves. Plants were inoculated with the mycelial blocks of *R. solani* 24 h post-*Rs*CRZ1-dsRNA-LDH treatment. (**F**) Relative lesion sizes on maize leaves caused by *R. solani* at different times after *Rs*CRZ1-dsRNA-LDH spraying. (**G**) RT-qPCR detection of *RsCRZ1* mRNA in *R. solani* in infected maize leaves at different time points after treatment with *Rs*CRZ1-dsRNA or *Rs*CRZ1-dsRNA-LDH. Mycelial blocks of *R. solani* were inoculated 24 h after *Rs*CRZ1-dsRNA and *Rs*CRZ1-dsRNA-LDH were sprayed on the leaves. ns, no significant differences; *, *p* < 0.05; **, *p* < 0.01.

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
