# Peer review of "Nanosheet-Facilitated Spray Delivery of dsRNAs Represents a Potential Tool to Control Rhizoctonia solani Infection"

_ijms, 2022, doi:10.3390/ijms232112922_

Round 1
Reviewer 1 Report
1. Be specific with the pathogen. Is it the same subspecies that cause disease in maize and rice.
2. Provide more information on the role of selected genes and what was basic for selecting these three genes
3. Why weight of sclerotium was increased with RsCAT-dsRNA treatment.
4. For systemic movement, how far was distant tissue taken for assessment of mRNA level.
5. Disease control experiment was conducted with RsCRZ1-dsRNA not with other genes, any specific reason for selecting this gene.
6. What was ratio of LDH and dsRNA for disease control and how much time it was incubated before being sprayed on plant
Reviewer 2 Report
The submitted manuscript ijms-1961135 determined the effects of the double-stranded RNAs loaded on layered double hydroxide (LDH) nanosheets in controlling Rhizoctonia solani pathogen infection and development. It is an interesting and important study as those findings might help the scientific community to understand the applicability of this approach to control the mentioned pathogen infection. The manuscript has systematically tested this technology, but, as pointed out by the authors, it would be interesting to know the effect of combining three dsRNAs (PG1, CAT CRZ1) and/or together with LDH. Overall, the work appears to have been performed in an acceptable manner, although some minor improvements are desirable.
L46: "genes expression" change to "gene expression".
L93, Fig. 1B: Did the authors isolate protoplast from mock treatment? Should it not have a comparison image?
L125, Fig. 2B: Although the bar chart shows a significant difference between CK and RsCRZ1-dsRNA, the image for 0.1 M and 0.2 M Ca2+ seems to show no differences. This might also be similar to Fig. 3D. Also, please label what are CK and MM in the figure caption.
L165-166: Wondering why tobacco? Perhaps, the authors could justify it.
L189-190: Should it be "dsRNA-EGFP" or "EGFP-dsRNA"? Perhaps, the authors could standardize it.
L222: How was 5% estimated?
L228, Fig. 5B: The gel did not include control?
L335: Suggest the authors include information about the replicates. Only section 4.9 describes this information in the current version. Also, does "three replications" mean three leaves? Or a few leaves, but the experiment was repeated three times?
L338: The molecular biological identification of the strain probably needs further description.
L342: Please define what is "nt' when the first time mentioned, i.e., nucleotides.
L350: “R. Solani” change to “R. solani”.
L363, 376 & 377: The text's font size seems incorrect.
L371-372: Please rephrase this sentence, i.e., "Observe the growth…".
L425: How the authors obtained LDH? Synthesize?
Round 2
Reviewer 2 Report
The authors have addressed my comments. Well done.